# Secular Trends in Physical Fitness of Peruvian Children Living at High-Altitude

**DOI:** 10.3390/ijerph20043236

**Published:** 2023-02-12

**Authors:** Carla Santos, Sara Pereira, Olga Vasconcelos, Go Tani, Donald Hedeker, Peter T. Katzmarzyk, José Maia, Alcibíades Bustamante

**Affiliations:** 1Centre of Research, Education, Innovation and Intervention in Sport (CIFI2D), Faculty of Sport, University of Porto, 4200-450 Porto, Portugal; 2Research Center in Sport, Physical Education, and Exercise and Health (CIDEFES), Faculty of Physical Education and Sports, Lusófona University, 1749-024 Lisboa, Portugal; 3Motor Behavior Laboratory, School of Physical Education and Sports, University of São Paulo, São Paulo 05508-030, Brazil; 4Department of Public Health Sciences, University of Chicago, Chicago, IL 60007, USA; 5Pennington Biomedical Research Center, Baton Rouge, LA 70808, USA; 6School of Physical Education and Sports, National University of Education Enrique Guzmán y Valle, 60637 La Cantuta, Lurigancho-Chosica 15472, Peru

**Keywords:** secular trends, physical fitness, children, high-altitude, Peru

## Abstract

The physical fitness of children is an important marker of health, and monitoring its temporal changes provides important information for developing interventions. We aimed to: (1) describe secular trends in physical fitness across age, within each sex, in Peruvian schoolchildren; and (2) verify if these trends are also present when adjusted for changes in height and weight. We sampled 1590 children (707 in 2009; 883 in 2019), aged 6–11 years. Physical fitness was assessed with four tests from the EUROFIT battery. ANOVA and ANCOVA statistical models were used. Results showed that with increasing age, girls and boys were significantly stronger in all PF tests, except for the case of flexibility in girls. In 2019, girls were stronger (handgrip) and more flexible than in 2009; but lower values were evident in standing long jumps in both sexes. Age-by-year interactions were statistically significant for agility in both sexes, with significant differences occurring at different ages. These trends did not change when adjusted for temporal changes in height and weight. Our research provides important data for local governments to implement public policies and practices to improve physical fitness levels in children.

## 1. Introduction

Societal and ecological changes across time lead to changes in the way children grow and develop, which can have significant health impacts [1]. In recent decades, several Peruvian regions have experienced demographic, socioeconomic, and cultural changes that have disrupted long-established ways of life [2,3]. For example, within a decade (from 2000 to 2010), public funds progressively increased [4], and programs to combat children’s malnutrition also significantly improved [5]. This is a result of rapid urbanization, industrialization, and advances in public health and safety that have occurred in developed countries throughout the years. These rapid economic advances have also had impacts in the Huancayo province in the city of Junín, located on a plateau at 4107 m [6]. For example, in the last decade, between 2009 to 2019, socio-economic conditions in Junín improved, with the Human Development Index (HDI) increasing from 0.42 to 0.49, and per capita family income increasing from 434.15 to 707.32 (SOL per month) [7]. Furthermore, a decrease in the stunted physical growth of children has also been reported, which reflects positive changes in economic conditions and their correlates—better nutrition, health, and quality of life [8]. Although available research indicates that most of the socio-economic indicators are heading in a constructive direction, based on an optimistic long-term vision, the effects of such contextual changes on physical fitness (PF) in Peruvian school children living at high altitude have not been studied.

PF, as a multifaceted health marker, is known to play a role in children’s growth and development [9]. It is often described as “a state of being that reflects a person’s ability to perform specific exercises or functions and is related to present and future outcomes” [10]. It has been shown that inferior levels of PF from early ages may have undesirable health consequences, namely a higher incidence of metabolic risk factors and poorer cardiovascular health profiles during adolescence and adulthood [11]. Furthermore, it has been suggested that childhood is a critical time window for PF development which is linked to motor skill proficiency [12]. In this time window, children are expected to acquire proficiency in locomotor and object control skills which will allow them to learn new skilled actions in response to the demands of different contexts—from school to their daily chores [13,14]. We contend that by studying temporal changes in children’s PF levels, relevant information can be provided to socio-political and educational systems to promote adequate and efficient intervention programs in Peru to enhance children’s health.

There is a common perception that PF among school children has declined in recent generations. This trend has been explained by increases in sedentary behaviors, lack of habitual moderate-to-vigorous physical activity, and the augmented accessibility to energy-rich food [15]. In fact, population-based studies [16,17,18], as well as recent systematic reviews [19,20,21,22], corroborated this perception, relating a negative secular trend in the PF of children since the mid-20th century. However, there are also contrasting results related to some PF components. For example, Masanovic, et al. [20] revealed a constant decline in muscle strength in children and adolescents from 14 countries (China, Finland, Sweden, Belgium, New Zealand, Denmark, Spain, Norway, Mozambique, Poland, USA, Lithuania, Portugal, and Canada) between 1969 and 2017. Conversely, Dooley et al. [22] reported a progressive increase in handgrip strength between 1967 and 2017 in children and adolescents from thirteen high-income countries, five upper-middle-income countries, and one low-income country. Dos Santos et al. [18] showed a negative secular trend across three time points (1992, 1999, 2012) in the speed/agility, flexibility, and cardiorespiratory fitness Mozambican boys and girls, whereas a positive trend was observed (only in upper-body strength and coordination/agility) from 1999 to 2014 in young Croatians [23]. These differences among studies and regions of the world likely reflect different secular trends in macro-level societal factors.

Notwithstanding the available body of data regarding secular trends in PF, there is limited information on children living in developing countries [24]. This gap is more noticeable among indigenous Latin American populations such as Peruvians, who live in rather challenging conditions, namely at high altitude, which leads to associated factors such as increased solar radiation, decreased ambient oxygen tension, extreme diurnal ranges in temperature, arid climate, and poor soil quality. The present study was designed to investigate secular trends in PF in 6–11-year-old Peruvian boys and girls between 2009 and 2019 living at high altitude. Specifically, we intended to: (1) describe secular trends in PF across age and sex in Peruvian primary school children; and (2) verify if secular trends were also present when adjusted for temporal trends in both height and weight.

## 2. Materials and Methods

### 2.1. Design and Participants

The data for the present article comes from two major research projects: the first, a cross-sectional study titled “The Peruvian Health and Optimist Growth Study” which was carried out between 2009 and 2010, probing the relationships between physical growth, motor development and health in Peruvian children and adolescents, as well as their families [25]; the second, named “Growth, Motor Performance and Lifestyles in Peruvian Schoolchildren Living at Different Altitudes. A Mixed-Longitudinal Study” is an ongoing project that started in 2019 and its main aims are to investigate the relationships between physical growth, motor performance, and lifestyles of children and adolescents living at sea level and at high-altitude, as well as the dynamics and stability of their processes over four years based on a mixed-longitudinal study. The two cohorts of subjects, both boys and girls, from 2009 and 2019 reside at high-altitude in the city of Junín (4107 m above sea-level). In total, the sample comprises 1590 primary school children (349 females and 359 males in 2009; 456 females and 433 males in 2019), aged 6–11 years (Table 1). Only children with complete data on all study variables as well as with signed informed consent were included in the present report. The Ethics Committee of the School of Physical Education and Sports, National University of Education Enrique Guzmán y Valle, Peru (UNE EGyV) approved both projects.

### 2.2. The City of Junín

Junín is the capital of Junín province. It is located on a plateau at 4107 m on the southern shore of Lake Junín or Chinchaycocha and is characterized by a very heterogeneous topography. Located in deep valleys and protected by mountains, the climate in the region is characterized by very cold winters and dry summers. The average annual temperature is 12 °C, the maximum high is 17 °C and the minimum low is 0 °C. Here, the residents deal with hypoxic stress, low temperatures and relative humidity, high cosmic radiation, and fewer hours of sunlight and twilight. The main livelihood is stockbreeding. Geographic, demographic, socioeconomic, sportive, and physical education classes characteristics in the Junín region are provided in Table 2.

### 2.3. Measurements and Tests

#### 2.3.1. Physical Fitness

Physical fitness was assessed using four standardized tests from the EUROFIT battery [26]: (1) handgrip strength—using a hand-held dynamometer (Takei Hand Grip Dynamometer^®^, Takei Scientific Instruments Co., Ltd., Tokyo, Japan), and the maximum static strength (kg^f^) was measured with the dominant hand; (2) standing long jump—all children jumped as far as possible landing on both feet without falling backwards, and the maximum horizontal distance attained was measured in centimeters; (3) sit and reach—all children were seated on the floor with their legs extended to front, and subjects reached forward, and the maximum distance achieved with the tip of the middle fingers through trunk flexion was measured using a standardized wooden stand; (4) shuttle-run agility test (10 × 5 m). The marker cones were positioned 5 m apart, and when instructed by the timer, children ran to the opposite marker, turned, and returned to the starting line. This was repeated 5 times without stopping (covering 50 m total), and time in seconds was used as the outcome.

#### 2.3.2. Anthropometry

Using standardized protocols [27], standing height was measured with the child´s head positioned in the Frankfurt plane using a portable stadiometer (Sanny, Model ES-2060, São Paulo, Brazil) to the nearest 0.1 cm. Furthermore, using a digital scale (Pesacon, Model IP68, Lima, Peru) body mass was measured to the nearest 0.1 kg.

### 2.4. Data Quality Control

Data quality control was assured through various stages. Firstly, the principal investigator (AB) trained all team members on all measurement procedures. Secondly, using a sample of 50 children from each cohort (2009 and 2019), checks were systematically undertaken in terms of data collection, data entry, and data processing as well as estimating reliability. Thirdly, a random sample of 4–5 children were daily reassessed for in-field reliability. In both cohorts, the intra-observer technical errors of measurement were: 2009, height = 0.13 cm and weight = 0.18 kg; 2019, height = 0.27 cm and weight = 0.48 kg. Furthermore, ANOVA-based intraclass correlation coefficient for PF tests in 2009 were: R = 0.96 (handgrip), R = 0.90 (standing long jump), R = 0.97 (sit and reach), and R = 0.66 (shuttle-run). In 2019, R = 0.98 (handgrip), R = 0.95 (standing long jump), R = 0.99 (sit and reach), and R = 0.80 (shuttle-run).

### 2.5. Statistical Procedures

Means and standard deviations were used as descriptive statistics. Initially, a two-factor ANOVA model was used to compare differences, within each sex, for each PF test between 2009 and 2019 by age, as well as age-by-study year (interaction). If the results were statistically significant, then a two-factor ANCOVA model was used with height and weight as covariates. Similar to the previous ANOVA model, ANCOVA was also run independently for each sex. Then, we contrasted boys or girls, across study year within each age. Effect sizes were also calculated [partial eta squared (η^2^)]. Stata 17 software [28] was used in all analyses, especially the *Margins and Marginsplot* modules. The level of significance level was 5%.

## 3. Results

Table 3 and Figure 1 show results from the two-factor ANOVA for each PF test in girls between the 2009 and 2019 cohorts. With increasing age, girls were stronger in handgrip (F = 156.06, *p* < 0.001, η^2^ = 0.497), with more explosive lower body strength (F = 41.91, *p* < 0.001, η^2^ = 0.210), and were more agile (F = 20.01, *p* < 0.001, η^2^ = 0.112), but not more flexible (F = 1.84, *p* = 0.103, η^2^ = 0.012). When compared to 2009, girls in 2019 performed better in handgrip (F = 10.33, *p* < 0.001, η^2^ = 0.012) and sit and reach (F = 38.76, *p* < 0.001, η^2^ = 0.047), but performance was worse in standing long jump (F = 38.77, *p* < 0.001, η^2^ = 0.047). This trend is not evident in agility (F = 1.52, *p* = 0.217, η^2^ = 0.002). Yet, the age-by-year interactions proved to be statistically significant only for agility (F = 8.77, *p* < 0.001, η^2^ = 0.528), with differences occurring at 6 years (favoring children from 2009 cohort study), and 10 and 11 years (favoring children from 2019 cohort study).

After adjusting for trends in height and weight (ANCOVA) a similar tendency was observed as in the previous analysis, but in handgrip no difference was found between cohorts (F = 1.79, *p* = 0.181, η^2^ = 0.002) (please see Appendix A and Appendix A).

Table 4 and Figure 2 show results from the two-factor ANOVA for each PF test in boys between 2009 and 2019 cohorts. With increasing age, boys were stronger in handgrip (F = 118.88, *p* < 0.001, η^2^ = 0.433), with more explosive lower body strength (F = 62.02, *p* < 0.001, η^2^ = 0.285), they were more agile (F = 15.67, *p* < 0.001, η^2^ = 0.091), and they were more flexible (F = 8.35, *p* < 0.001, η^2^ = 0.051). When compared to 2009, boys in 2019 only underperformed compared with their peers in standing long jump (F = 23.40, *p* < 0.001, η^2^ = 0.029). This trend is not evident in the other PF tests (handgrip: F = 2.70, *p* = 0.101, η^2^ = 0.003; shuttle-run: F = 0.45, *p* = 0.504, η^2^ = 0.001; sit and reach: F = 0.08, *p* = 0.775, η^2^ = 0.001). Yet, the age-by-year interactions proved to be statistically significant only for agility F = 5.13, *p* < 0.001, η^2^ = 0.032), with differences occurring at 6 and 7 years (favoring children from 2009 cohort study), and 11 years (favoring children from 2019 cohort study).

After adjusting for trends in height and weight (ANCOVA), a similar tendency in results was observed as with the previous analysis. Yet, in the shuttle-run test no significant differences were found across ages (F = 2.10, *p* = 0.063, η^2^ = 0.013) with body size adjustments (please see Appendix A and Appendix A).

## 4. Discussion

This study provides novel results regarding secular trends in PF among children living in the Peruvian mountains. It also identifies tendencies when adjusted for temporal trends in both height and weight.

The results showed that, as expected, with increasing age, Peruvian children of both sexes had significantly higher PF levels, i.e., becoming stronger in handgrip, with more explosive lower body strength, and more agility and flexibility (only in boys). During childhood, children grow and develop, experiencing significant changes in their body shape and composition, becoming taller, heavier, and stronger [29]. Concomitantly, as their bodies develop, they also acquire, and refine a wide range of fundamental motor skills which enables them to perform more complex movements. They learn to run, jump, catch and throw, as well as combine different fundamental motor skills, which can contribute to improve their performance in PF assessments [30]. In fact, children follow the same biological “rule” in the process of developing their fundamental motor skills—a gradual development, followed by an internalizing process and later by improving them. However, it is important to note that continued practice and instruction are necessary to increase children’s motor repertoire as they increase their age.

Our results revealed significant trends in Peruvian children’s PF levels over the last decade. In general, 2019 girls were stronger (handgrip) and more flexible, but lower scores were evident in standing long jump in both sexes. Furthermore, no changes were found in the other PF tests across the analyzed period. The most relevant outcome of the present study is the substantive increase from 2009 to 2019 in some PF components in both sexes, in contrast to the general trends suggesting that children PF tends to decline in various countries in the previous three decades, given the development of unhealthy lifestyle behaviors from an early age [31]. In line with our results, Smpokos, et al. [32] also described positive secular changes in Greek boys and girls (1992/1993–2006/2007) in their aerobic capacity, anaerobic strength, and flexibility. Likewise, Moliner-Urdiales, et al. [33] also found that Spanish adolescents of both sexes in 2006/2007 demonstrated improved performance in a cardiovascular endurance test and in speed agility compared to their peers from five years prior (2001–2002). A positive trend was also observed in upper-body strength and coordination/agility from 1999 to 2014 in young Croatians [23]. A meta-analysis using data from 14 countries also revealed that in several studies, the handgrip strength test showed a growth trend in strength for girls [34,35]. This trend may be in accordance not only with the hypothesis that muscular strength is proportional to its cross-sectional [36] but also with the fact that stature has increased over the last 40 years [37]. Positive worldwide trends in flexibility levels have also been reported, for example, in Canada [34], based on intervention programmes with additional exercises. On the contrary, we are yet to know, precisely, the explanation for trends observed in New Zealand [38] and Mozambique [18].

Another interesting result of our study was the age-by-year interactions which was statistically significant only for agility in both sexes, with differences occurring at different ages, favoring 2009 girls at 6 years and boys at 6 and 7 years; on the contrary, it also favors 2019 girls at 10 and 11 years, and 11 years old boys. Similar age-by-year discrepancies were reported by Matton, et al. [39], showing a declining trend in a boys (12–18 years old) shuttle run test over 35 years; on the contrary, in girls, a slight decrease was seen at 14 years. Furthermore, previous studies [15,16] on children’s secular trends in PF did not report age-by-year interactions which poses problems when comparing our results. In the context of Junín, we speculate that these results can be explained by biological factors as well as contextual temporal changes. Firstly, it is important to note that during childhood, different physical growth phases may occur during the mid-growth spurt, namely in terms of timing and tempo. For example, Pereira, et al. [40], using a mixed-longitudinal study in a Portuguese sample, revealed that boys’ and girls’ age-at-peak of their mid-growth spurt in stature occurred at 7.8 ± 0.47 years and 8.0 ± 0.72 years. Then, they identified the timing, intensity, and sequences of PF spurts aligned by the age-at-peak of their mid-growth spurt and concluded that agility spurt occurred before age-at-peak in girls and after in boys. These different timings apparently coincide with the highest differences in agility performance in specific age groups. This will probably help to better grasp changes in Peruvian children’s performance outcomes growth spurts. Moreover, local governments also invested in education policies and practices during the last decade. For example, in 2009, physical education classes in primary schools only had, on average, a duration of two hours a week, while a decade later the Peruvian Ministry of Education approved an extra hour per week, which may help explain differences in the agility outcomes, as a greater duration of physical education classes is expected to be linked with higher gains of PF capacities over time, especially after 8 years onwards (which may be also linked to age at peak height in the mid-growth spurt). Possible effects of improvement in regional socioeconomic conditions and food security may also explain the results, although we have no data available, as mentioned in the limitations. Care should be taken when dealing with agility secular trends because available reports use different tests.

Finally, when dealing with PF secular trends, a special care is required concerning the interpretation of the results, because changes in body size, as a result of growth and maturation, are expected to also occur [41]. Bearing this in mind, we adjusted secular trends for temporal trends in both children’s height and weight (ANCOVA model). Our results revealed a similar tendency as in the previous analysis (ANOVA model), except in handgrip, in which no difference was found in boys between cohorts, as well as in the shuttle-run test, with no significant differences in girls across the period. In general, this null change in most PF tests may be explained, in part, by the overall maintenance of height and weight between 2009 and 2019 (please see Appendix A, where we also edited the BMIz-score). Nevertheless, exceptions were found in handgrip, since the dimensions of body size and body mass, normally favoring boys, directly influence muscular strength [42]. In turn, height and weight values are also correlated with the capacity children usually display in rapidly changing the center of mass of their body vertically and horizontally as required in the agility test [43].

It is important to acknowledge the limitations of this study. Firstly, no adjustments were made for putative influences of physical activity levels and nutrition habits. This information was not available because of funding limitations. Secondly, we did not consider the family income and/or the socioeconomic status of the children from both study cohorts. In fact, all children are from public schools, and there is apparently no specific ground to test for a putative effect of these conditions in PF secular trends. In any case, future studies should consider these variables to reveal a clearer picture of the correlates of the observed PF secular trends. This study has also important aspects that need to be acknowledged. Firstly, this is the first time that PF secular trends of Peruvian children living at high-altitude are described using a relatively large and representative sample. Secondly, our data cover a decade (2009–2019) of socioeconomic changes in Junín region. Thirdly, data collection relied on objective and reliable PF tests, and rigorous quality control measures were carefully applied. Finally, it with would be of great interest to investigate putative changes in these children after the COVID-19 pandemic.

## 5. Conclusions

In conclusion, the present study showed that with increasing age, Peruvian children of both sexes significantly improved their PF levels (except for flexibility in girls). A significant secular trend was also revealed, with girls in 2019 being stronger in handgrip and flexibility. In turn, girls and boys in 2019 underperformed in standing long jump compared with their 2009 peers. Results also indicated that age-by-year interactions were statistically significant only for agility in both sexes, with significant differences occurring at different ages. Finally, after adjusting for trends in height and weight, a similar tendency in results was observed as previous analysis, except in handgrip, in which no difference was found in boys between cohorts, as well as in the shuttle-run test, with no significant differences in girls across ages.

Altogether, our results provide important information about temporal changes in children’s PF levels living in the Peruvian mountains, which may have important practical implications. On the one hand, it can help regional political governments to design new approaches for reintegration and local contextual development, as well as to implement effective public education policies and intervention strategies to improve PF levels. On the other hand, it can help physical education teachers and sports coaches to outline innovative intervention programs to improve children’s PF levels.

## Figures and Tables

**Figure 1 ijerph-20-03236-f001:**
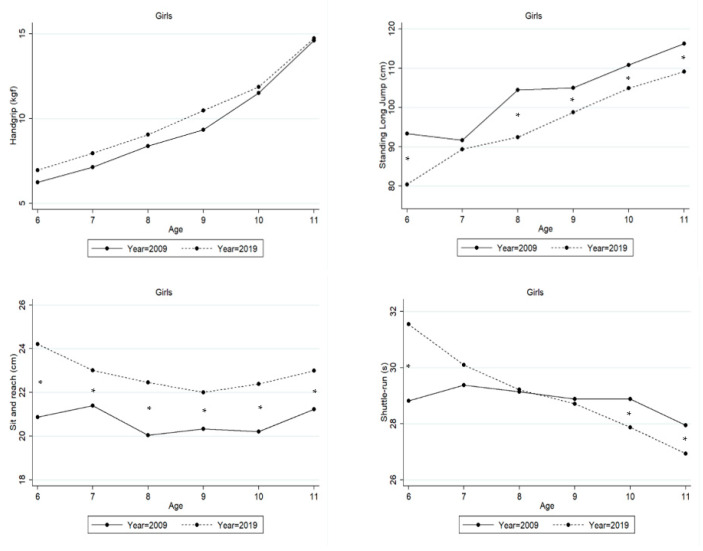
Means of Peruvian girls’ physical fitness between 2009 and 2019. Significant differences are marked by *.

**Figure 2 ijerph-20-03236-f002:**
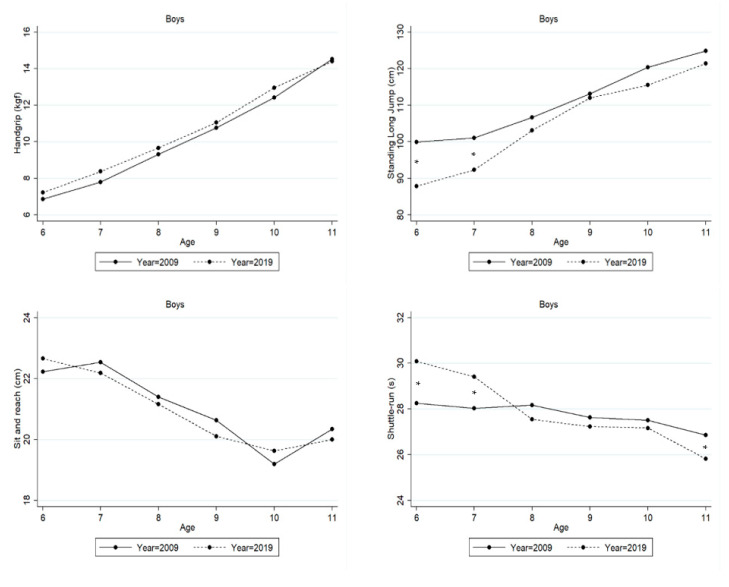
Means of Peruvian boys’ physical fitness between 2009 and 2019. Significant differences are marked by *.

**Table 1 ijerph-20-03236-t001:** Sample size by sex, age, and study cohort.

Age *	2009			2019		
(Years)	Girls	Boys	Total	Girls	Boys	Total
6	38	48	86	50	50	100
7	55	47	102	72	72	144
8	47	62	109	75	88	163
9	66	74	140	78	84	162
10	69	70	139	89	71	160
11	74	57	131	87	67	154
TOTAL	349	358	707	451	432	883

* Children aged 6.00 to 6.99 years were considered as 6, children aged 7.00 to 7.99 years as 7, and so on till 11 years.

**Table 2 ijerph-20-03236-t002:** Geographic, demographic, socioeconomic, sportive, and physical education classes characteristics in Junín city in 2009 and 2019.

Characteristics	2009	2019
Geographic		
Geographical area (km^2^)	883.8	
**Demographic**		
Total population (inhabitants)	11,242	6833
**Socioeconomic**		
Human Development Index	0.42	0.49
Per capita family income (SOL per month)	434.15	707.32
Primary production	Stockbreeding/Agriculture	
**Sportive features**		
Number of clubs	12	08
Number of practitioners	260	150
Sports practiced	Soccer, volleyball, athletics, basketball	Soccer, volleyball, athletics, basketball, handball
**Infrastructure for physical activity and sports**		
Parks	Yes	Yes
Kids’ playground	No	Yes
Pool	No	No
Multisport indoor	Yes	Yes
Multisport outdoor	Yes	Yes
Gymnastics complex	No	No
**Physical Education (PE) classes**		
Total population (schoolchildren)	3560	2649
PE frequency (p/week)	1	1
PE duration (hours)	Primary and Secondary (2 h)	Primary (3 h); Secondary (2 h)
Equipment available (type)	Balls, cones, chess game	Balls, cones, chess game, gymnastic mats, banks
Extracurricular activities (type)	Folk dances	Folk dances, rhythmic gymnastics
Number of participants in the school sports program	450	400

**Table 3 ijerph-20-03236-t003:** Descriptive statistics (means ± standard errors), differences (diff), and their corresponding 95% confidence intervals (95%CI) for each physical fitness test in girls between 2009 and 2019 cohorts.

Girls																
	Handgrip (kg^f^)				Standing Long Jump (cm)				Sit and Reach (cm)				Shuttle-Run (s)			
Age (y)	2009	2019	Diff.	95%CI	2009	2019	Diff.	95%CI	2009	2019	Diff.	95%CI	2009	2019	Diff.	95%CI
6	6.2 ± 0.4	7.0 ± 0.4	0.8 ^ns^	−0.44; 1.86	93.3 ± 2.8	80.4 ± 2.4	−12.9 **	−20.08; −5.69	20.9 ± 0.8	24.2 ± 0.7	3.3 **	1.32; 5.35	28.8 ± 0.4	31.5 ± 0.3	2.7 **	1.68; 3.77
7	7.1 ± 0.4	8.0 ± 0.3	0.9 ^ns^	−0.13; 1.78	91.7 ± 2.3	89.4 ± 2.0	−2.3 ^ns^	−8.25; 3.73	21.3 ± 0.6	23.0 ± 0.6	1.7 *	−0.06; 3.29	29.4 ± 0.3	30.1 ± 0.3	0.7 ^ns^	−0.15; 1.59
8	8.4 ± 0.4	9.1 ± 0.3	0.7 ^ns^	−0.32; 1.67	104.5 ± 2.5	92.4 ± 2.0	−12.1 **	−18.26; −5.82	20.0 ± 0.7	22.5 ± 0.6	2.5 **	0.67; 4.15	29.1 ± 0.4	29.2 ± 0.3	0.1 ^ns^	−0.83; 0.97
9	9.3 ± 0.3	10.5 ± 0.3	1.2 ^ns^	0.24; 2.02	105.0 ± 2.1	98.8 ± 1.9	−6.2 *	−11.87; −0.68	20.3 ± 0.6	22.0 ± 0.5	1.7 *	0.11; 3.24	28.9 ± 0.3	28.7 ± 0.3	−0.2 ^ns^	−0.97; 0.65
10	11.5 ± 0.3	11.9 ± 0.3	0.4 ^ns^	−0.51; 1.20	110.8 ± 2.0	104.9 ± 1.8	−5.9 *	−11.25; −0.52	20.2 ± 0.6	22.4 ± 0.5	2.2 **	0.68; 3.68	28.9 ± 0.3	27.9 ± 2.3	−1.0 **	−1.79; −0.23
11	14.6 ± 0.3	14.7 ± 0.3	0.1 ^ns^	−0.71; 0.98	116.2 ± 2.0	109.1 ± 1.8	−7.1 **	−12.39; −1.82	21.2 ± 0.5	23.0 ± 0.5	1.8 *	0.28; 3.24	27.9 ± 2.4	26.9 ± 2.4	−1.0 **	−1.78; −0.24
Two-Factor ANOVA Model Results																
Age	F = 156.06, *p* < 0.001, η^2^ = 0.497				F = 41.91, *p* < 0.001, η^2^ = 0.210				F = 1.84, *p* = 0.103, η^2^ = 0.012				F = 20.01, *p* < 0.001, η^2^ = 0.112			
Year	F = 10.33, *p* < 0.001, η^2^ = 0.012				F = 38.77, *p* < 0.001, η^2^ = 0.047				F = 38.76, *p* < 0.001, η^2^ = 0.047				F = 1.52, *p* = 0.217, η^2^ = 0.002			
Age-by-Year	F = 0.63, *p* = 0.679, η^2^ = 0.003				F = 1.54, *p* = 0.175, η^2^ = 0.009				F = 0.48, *p* = 0.792, η^2^ = 0.003				F = 8.46, *p* < 0.001, η^2^ = 0.509			

^ns^ = non-statistically significant; *, *p* < 0.05; **, *p* ≤ 0.01.

**Table 4 ijerph-20-03236-t004:** Descriptive statistics (means ± standard errors), differences (diff) and their corresponding 95% confidence intervals (95%CI) for each physical fitness test in boys between 2009 and 2019 cohorts.

Boys																
	Handgrip (kg^f^)				Standing Long Jump (cm)				Sit and Reach (cm)				Shuttle-Run (s)			
Age (y)	2009	2019	Diff.	95%CI	2009	2019	Diff.	95%CI	2009	2019	Diff.	95%CI	2009	2019	Diff.	95%CI
6	6.9 ± 0.4	7.2 ± 0.4	0.3 ^ns^	−0.75; 1.45	99.9 ± 2.3	87.9 ± 2.3	−12.0 **	−18.47; −5.74	22.2 ± 0.7	22.7 ± 0.7	0.5 ^ns^	−1.42; 2.30	28.3 ± 0.4	30.1 ± 0.4	1.8 **	0.74; 2.90
7	7.8 ± 0.4	8.4 ± 0.3	0.6 ^ns^	−0.44; 1.61	101.1 ± 2.3	92.3 ± 1.9	−8.8 **	−14.67; −2.85	22.5 ± 0.7	22.2 ± 0.6	−0.3 ^ns^	−2.07; 1.37	28.0 ± 0.4	29.4 ± 0.3	1.4 **	0.36; 2.38
8	9.3 ± 0.3	9.7 ± 0.3	0.4 ^ns^	−0.55; 1.26	106.7 ± 2.0	103.1 ± 1.7	−3.6 ^ns^	−8.76; 1.69	21.4 ± 0.6	21.2 ± 0.5	−0.2 ^ns^	−1.76; 1.29	28.2 ± 0.3	27.5 ± 0.3	−0.7 ^ns^	−1.52; 0.27
9	10.8 ± 0.3	11.0 ± 0.3	0.2 ^ns^	−0.59; 1.15	113.1 ± 1.9	112.0 ± 1.8	−1.1 ^ns^	−6.14; 3.90	20.6 ± 0.5	20.1 ± 0.5	−0.5 ^ns^	−1.98; 0.95	27.6 ± 0.3	27.2 ± 0.3	−0.4 ^ns^	−1.26; 0.46
10	12.4 ± 0.3	13.0 ± 0.3	0.6 ^ns^	−0.38; 1.45	120.4 ± 1.9	115.5 ± 1.9	−4.9 ^ns^	−10.17; 0.44	19.2 ± 0.6	19.6 ± 0.6	0.4 ^ns^	−1.11; 1.99	27.5 ± 0.3	27.2 ± 0.3	−0.3 ^ns^	−1.24; 0.57
11	14.5 ± 0.4	14.4 ± 0.3	−0.1 ^ns^	−1.10; 0.86	124.8 ± 2.1	121.4 ± 2.0	−3.4 ^ns^	−9.10; 2.25	20.3 ± 0.6	20.0 ± 0.6	−0.3 ^ns^	−2.01; 1.31	26.9 ± 0.4	25.8 ± 0.3	−1.1 *	−2.01; −0.07
Two-Factor ANOVA Model Results																
Age	F = 118.88, *p* < 0.001, η^2^ = 0.433				F = 62.02, *p* < 0.001, η^2^ = 0.285				F = 8.35, *p* < 0.001, η^2^ = 0.051				F = 15.67, *p* < 0.001, η^2^ = 0.091			
Year	F = 2.70, *p* = 0.101, η^2^ = 0.003				F = 23.40, *p* < 0.001, η^2^ = 0.029				F = 0.08, *p* = 0.775, η^2^ = 0.001				F = 0.45, *p* = 0.504, η^2^ = 0.001			
Age-by-Year	F = 0.25, *p* = 0.939, η^2^ = 0.002				F = 1.85, *p* = 0.101, η^2^ = 0.118				F = 0.26, *p* = 0.936, η^2^ = 0.002				F = 5.13, *p* < 0001, η^2^ = 0.032			

^ns^ = non-statistically significant; *, *p* < 0.05; **, *p* ≤ 0.01.

## Data Availability

The data are property of the School of Physical Education and Sports, National University of Education Enrique Guzmán y Valle, Peru (UNE EGyV), as is therefore protected from being freely shared.

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
