# Peer review of "Secular Trends in Physical Fitness of Peruvian Children Living at High-Altitude"

_ijerph, 2023, doi:10.3390/ijerph20043236_

Round 1

Reviewer 1 Report

Review for: MDPI IJEPRH
Secular Trends in Physical Fitness of Peruvian Children Living 2 at High-Altitude

1. What is the main question addressed by the research?

The study aims to: (1) describe secular trends in physical fitness across age, within each sex, in Peruvian schoolchildren; and (2) verify if these trends are also present when adjusted for changes in height and weight.

2. Do you consider the topic original or relevant to the field? Does it address a specific gap in the field?

This topic is really important because it reports data for local governments to implement public policies and practices to prove physical fitness levels in children but also to compare data of fitness worldwide.

3. What does it add to the subject area compared with other published material?

The data of fitness present in this paper is new and data is quite actual. There is limited information on children living in developing countries on secular trends in PF. This gap is more noticeable among indigenous Latin American populations, such as Peruvians, who live in rather challenging conditions, namely at high-altitude which is characterized by increased solar radiation, decreased ambient oxygen tension, extreme diurnal ranges in temperature, arid climate, and poor soil quality. During the last decades Peruvian regions experienced demographic, socioeconomic, and cultural changes that di rupted long-established ways of life. Although available research indicates that most of the social and economic indicators are moving in a positive direction, the effects of such contextual changes on physical fitness (PF) in Peruvian schoolchildren but also worldwide have not been studied or is rare.

4. What specific improvement should authors consider regarding the methodology? What further controls should be considered?

The study is well designed, the fitness test which was used is a validated tool (Eurofit) which is is widely published and used in many epidemiological studies. The staff was trained, a pilotstidy was set up and data control was conducted. Body hight and weigth were measured. For further studies, it is absolutely necessary that the physical activity and the social status are surveyed, but this is addressed by the authors themselves in the limitations.

5. Are the conclusions consistent with the evidence and arguments presented and do they address the main question posed?

Indication of the results in current state of research is comprehensible. The discussion of the limitations shows an adequat distance to the own findings. It would be good for the reader to know if other fitness surveys are planned that are particularly relevant with respect to impacts of the COVID pandemic. It would be nice if the authors could provide two sentences on this with a look into the future on how and if the study will be continued.

6. Are the references appropriate?

The references are appopriate and up-to-date, e.g. two actual reviews on the topic of secular trends are reported, I oly suggest to add one more:

Eberhardt, T., Niessner, C., Oriwol, D., Buchal, L., Worth, A., & Bös, K. (2020). Secular Trends in Physical Fitness of Children and Adolescents: A Review of Large-Scale Epidemiological Studies Published after 2006. International Journal of Environmental Research and Public Health, 17.

More references would be:

Ekblom, B., Engström, L. M., & Ekblom, Ö. (2007). Secular trends of physical fitness in Swedish adults. Scandinavian journal of medicine & science in sports, 17(3), 267-273. Shigaki, G. B., Batista, M. B., Paludo, A. C., Vignadelli, L. F. Z., Serassuelo Junior, H., & Ronque, E. R. V. (2019).

Secular trend of physical fitness indicators related to health in children. Journal of Human Growth and Development, 29(3), 381-389. Potočnik, Ž. L., Jurak, G., & Starc, G. (2020).

Secular trends of physical fitness in twenty-five birth cohorts of Slovenian children: A population-based study. Frontiers in public health, 8, 561273.

7. Please include any additional comments regarding tables.

The tables are easy to understand and follow, and most importantly, they provide data per age group that allows mean comparisons with other studies.

Overall Feedback:

ABSTRACT Abstract contains all important information of the paper

INTRODUCTION Introduction contains all important information, contains current and relevant literature and leads to the research question

METHODS The use of the methods is state of the art and the describtion of the methods is adequat.

RESULTS Successful and compact representation of the findings

DISCUSSION Indication of the results in current state of research is comprehensible. The discussion of the limitations shows an adequat distance to the own findings.

Great publication! Well designed study which provides important data for local governments to implement public policies and practices to prove physical fitness levels in children but also to compare data of fitness worldwide.

Author Response

Dear Petar Stojanovic,

Section Managing Editor

International Journal of Environmental Research and Public Health (IJERPH)

Firstly, we would like to thank you for the opportunity to submit a new version of our manuscript. Secondly, we want to acknowledge the generosity of the two reviewers for their critiques/comments/suggestions (in italics) that helped enhance the quality of the new draft. Please find below our answers. Changes in the new manuscript are marked in yellow.

Comments to the Authors

Reviewer 1

  1. What is the main question addressed by the research?

The study aims to: (1) describe secular trends in physical fitness across age, within each sex, in Peruvian schoolchildren; and (2) verify if these trends are also present when adjusted for changes in height and weight.

  1. Do you consider the topic original or relevant to the field? Does it address a specific gap in the field?

This topic is really important because it reports data for local governments to implement public policies and practices to prove physical fitness levels in children but also to compare data of fitness worldwide.

  1. What does it add to the subject area compared with other published material?

The data of fitness present in this paper is new and data is quite actual. There is limited information on children living in developing countries on secular trends in PF. This gap is more noticeable among indigenous Latin American populations, such as Peruvians, who live in rather challenging conditions, namely at high-altitude which is characterized by increased solar radiation, decreased ambient oxygen tension, extreme diurnal ranges in temperature, arid climate, and poor soil quality. During the last decades Peruvian regions experienced demographic, socioeconomic, and cultural changes that disrupted long-established ways of life. Although available research indicates that most of the social and economic indicators are moving in a positive direction, the effects of such contextual changes on physical fitness (PF) in Peruvian schoolchildren but also worldwide have not been studied or is rare.

  1. What specific improvement should authors consider regarding the methodology? What further controls should be considered?

The study is well designed, the fitness test which was used is a validated tool (Eurofit) which is widely published and used in many epidemiological studies. The staff was trained, a pilot study was set up and data control was conducted. Body height and weight were measured. For further studies, it is absolutely necessary that the physical activity and the social status are surveyed, but this is addressed by the authors themselves in the limitations.

  1. Are the conclusions consistent with the evidence and arguments presented and do they address the main question posed?

Indication of the results in current state of research is comprehensible. The discussion of the limitations shows an adequate distance to the own findings. It would be good for the reader to know if other fitness surveys are planned that are particularly relevant with respect to impacts of the COVID pandemic. It would be nice if the authors could provide two sentences on this with a look into the future on how and if the study will be continued.

  1. Are the references appropriate?

The references are appropriate and up-to-date, e.g. two actual reviews on the topic of secular trends are reported, I only suggest to add one more:

Eberhardt, T., Niessner, C., Oriwol, D., Buchal, L., Worth, A., & Bös, K. (2020). Secular Trends in Physical Fitness of Children and Adolescents: A Review of Large-Scale Epidemiological Studies Published after 2006. International Journal of Environmental Research and Public Health, 17.

More references would be:

Ekblom, B., Engström, L. M., & Ekblom, Ö. (2007). Secular trends of physical fitness in Swedish adults. Scandinavian journal of medicine & science in sports, 17(3), 267-273.

Shigaki, G. B., Batista, M. B., Paludo, A. C., Vignadelli, L. F. Z., Serassuelo Junior, H., & Ronque, E. R. V. (2019).

Secular trend of physical fitness indicators related to health in children. Journal of Human Growth and Development, 29(3), 381-389. Potočnik, Ž. L., Jurak, G., & Starc, G. (2020).

Secular trends of physical fitness in twenty-five birth cohorts of Slovenian children: A population-based study. Frontiers in public health, 8, 561273.

  1. Please include any additional comments regarding tables.

The tables are easy to understand and follow, and most importantly, they provide data per age group that allows mean comparisons with other studies.

Overall Feedback:

ABSTRACT Abstract contains all important information of the paper.

INTRODUCTION Introduction contains all important information, contains current and relevant literature and leads to the research question.

METHODS The use of the methods is state of the art and the description of the methods is adequate.

RESULTS Successful and compact representation of the findings.

DISCUSSION Indication of the results in current state of research is comprehensible. The discussion of the limitations shows an adequate distance to the own findings.

Great publication! Well-designed study which provides important data for local governments to implement public policies and practices to prove physical fitness levels in children but also to compare data of fitness worldwide.

Authors answer: We gratefully thank the reviewer for her/his considerate words and generous comments. The new references were added as suggested. Please note also that data has been collected prior and after the COVID-19 pandemic. These data sets are now under investigation by the team, and we will soon make this information available, hopefully.

Reviewer 2 Report

Thank you for the opportunity to review this manuscript entitled ‘Secular Trends in Physical Fitness of Peruvian Children Living at High-Altitude’ by Carla Santos and co-authors. It’s a relevant topic to investigate in different contexts and high-altitude is one of these. In the present study, the authors examined the evolution of physical fitness trends in Peruvian school-aged children. This study is interesting because physical fitness is a crucial marker of health, and this study shows new data about the trends of physical fitness in children who live in a high-altitude.

The manuscript is well-written and presents data that could be taken into account for publication in the International Journal of Environmental Research and Public Health after a full revision. In fact, there are many key points that require clarification.

Firstly: Please change the term “Secular trends” to “Temporal trends” both in the text and in the title since the period that your study took into consideration is 10 years.

Introduction: I think that this section is well-written, and it is focused on the objective of the study. I recommend only few suggestions.

Please change the term “substantiated” at line 70.

Please merge the authors of citation number 20 (line 73) and 18 (line 79) in the text and add only Masanovic et al. and Dos Santos et al.

Methods:  I suggest deleting the sentence from line 114 to line 116.

Although is admirable your precision in describing the city of Junín, I suggest presenting table 2 in the supplementary section, since a very good explanation of why it is important to analyze this city is provided in section 2.2

I suggest to describe the PF tests with more details with a subparagraph for each test, to make the methodology of this study reproducible in other studies. Also, I suggest to add reliability and IntraClass Correlation (ICC) for these tests.

Was the shuttle run a 10 x 5m or 5x5m? Because in the EUROFIT tests battery there is the 10x5m shuttle run, while reading your description I understood that you used a 5x5m shuttle run. Please clarify.

I suggest to show table S2 after table 1 in the method section and not in the supplementary.

Since BMIz-score is a crucial marker for health in childhood, I suggest adding the BMIz-score in table S2 and assessing the differences in BMIz-score between the two samples to strengthen your data.

Results: Please delete the term “significant” when reporting the results (for example line 203).

Please in the figure where there is a significant difference in PF between the age insert an appropriate symbol.

In line 219 the authors stated that the boys in 2019 were stronger than the boys in 2009. But in the table boys in 2019 jumped had a lower performance than their peers in 2009. This typo is also in lines 200-201 when reporting the results of the girls. Please correct.

In Tables 3 and 4 I suggest including a row above the others explaining the gender of subjects in the table.

Discussion: In the discussion session the authors described an increasing trend in PF abilities in both sexes, but the table presented in the results section shows a decline (for example the standing broad jump in boys decreased in all the ages while in the discussion you said the opposite (Line 271)). Accordingly, I suggest to modify the section of the discussion that compares these results with other studies.

Line 291–318 The explanation of the results obtained by the age-by-year analysis does not seem correlated with the results of the analysis. Specifically, you first stated that a possible explanation is due to the agility sprint, but since you compared children of the same age and found a difference between the two groups (2009-2019), your explanation is inadequate. Also, even if the increased hour of physical education in school could slightly improve the agility of children, this amelioration should have influenced children of all ages in the 2019 group and not only the performance of the 10 and 11 years old boys and girls.

Moreover, I suggest at lines 293-294 clarify these results since you have not compared boys and girls together. Specifically, you have reported that “favoring children from 2009 at 6 and 7

years” but as reported in table 3 and 4, you found a decreased performance for the shuttle run only in the 7 years old boys and not in girls. Then you reported “children from 2019 at 8, 10 and 11 years” but the improved performance at 10 years old was showed only in girls, while in any of both sexes was found an improved performance at the age of 8. Please be careful when describing your results.

Finally, I suggest adding a brief section explaining the possible positive/negative repercussion of these trends on children’s health and how your results could be helpful for Junìn physical education teachers and trainers (for example focusing their activities on the decreased physical fitness abilities).

Conclusions: For the same reasons stated in the discussion section please rephrase this paragraph.

Abstract: For the same reasons stated for the discussion section, I suggest updating this paragraph and add the correct interpretation of the results.

Author Response

Dear Petar Stojanovic,

Section Managing Editor

International Journal of Environmental Research and Public Health (IJERPH)

Firstly, we would like to thank you for the opportunity to submit a new version of our manuscript. Secondly, we want to acknowledge the generosity of the two reviewers for their critiques/comments/suggestions (in italics) that helped enhance the quality of the new draft. Please find below our answers. Changes in the new manuscript are marked in yellow.

Comments to the Authors

Reviewer 2

Firstly: Please change the term “Secular trends” to “Temporal trends” both in the text and in the title since the period that your study took into consideration is 10 years.

Authors answer: Before we start answering the reviewer, we would like to thank her/him for all the questions/comments/suggestions made. They will certainly improve the quality of the new draft hoping that it will now meets IJERPH standards.

Please note that although the expression secular trend may imply 100 years, in fact it can cover even from 6 years, 10 years, …, till 100 years. Here are a few examples of papers using Secular Trend in their titles and using different time-lags:

  • Shigaki et al (2019) study uses 3 data waves: 2002; 2005; 2010-2011. The first time-lag is 3 years, and the second is 5-6 years.
  • Costa et al (2017) report uses four quinquennials (1993-1998; 1998-2003; 2003-2008; 2008-2013).
  • Andersen et al (2009) study covers three different years - 1983, 1997, and 2003; the first time-lag is 14 years, and the second is 6 years.
  • Ekblom et al (2007) study covers a time lag of only 10 years.
  • Kasovic et al (2021) report relies on a time-lag of 15 years.
  • Fuhner et al (2020) update on secular trend in physical fitness of children and adolescents shows a variety of time-lags in published secular trend papers (from a time-lag of 6 years till a time-lag of 35 years).
  • Finally, in the monograph Pediatric Fitness. Secular trends and geographic variability edited by Tomkinson and Olds (2007) different authors also use a variable range of time-lags in their reports.

In sum, we hope the reviewer will not disagree with us in keeping the expression Secular Trend in the title instead of temporal trends.

References:       

Andersen LB, Froberg K, Kristensen PL, Moller NC, Resaland GK, Anderssen SA. Secular trends in physical fitness in Danish adolescents. Scand J Med Sci Sports. 2010 Oct;20(5):757-63. doi: 10.1111/j.1600-0838.2009.00936.x.

Costa AM, Costa MJ, Reis AA, Ferreira S, Martins J, Pereira A. Secular Trends in Anthropometrics and Physical Fitness of Young Portuguese School-Aged Children. Acta Med Port. 2017 Feb 27;30(2):108-114. doi: 10.20344/amp.7712.

Ekblom B, Engström LM, Ekblom O. Secular trends of physical fitness in Swedish adults. Scand J Med Sci Sports. 2007 Jun;17(3):267-73. doi: 10.1111/j.1600-0838.2006.00531.x.

Fühner T, Kliegl R, Arntz F, Kriemler S, Granacher U. An Update on Secular Trends in Physical Fitness of Children and Adolescents from 1972 to 2015: A Systematic Review. Sports Med. 2021 Feb;51(2):303-320. doi: 10.1007/s40279-020-01373-x.

Kasović M, Štefan L, Petrić V. Secular trends in health-related physical fitness among 11-14-year-old Croatian children and adolescents from 1999 to 2014. Sci Rep. 2021 May 26;11(1):11039. doi: 10.1038/s41598-021-90745-y.

Shigaki, G. B., Batista, M. B., Paludo, A. C., Vignadelli, L. F. Z., Serassuelo Junior, H., & Ronque, E. R. V. (2019). Secular trend of physical fitness indicators related to health in children. Journal of Human Growth and Development, 29(3), 381-389. doi: 10.7322/jhgd.v29.9537

Tomkinson GR, Olds TS (eds) (2007). Pediatric fitness. Secular trends and geographic variability. Medicine and Sport Sciences. (Vol. 50, pp. 46-66). Karger Publishers.

Introduction: I think that this section is well-written, and it is focused on the objective of the study. I recommend only few suggestions.

Please change the term “substantiated” at line 70.

Authors answer: Many thanks for your generous words. The word substantiated was changed.

Please merge the authors of citation number 20 (line 73) and 18 (line 79) in the text and add only Masanovic et al. and Dos Santos et al.

Authors answer: Done as suggested.

Methods:  I suggest deleting the sentence from line 114 to line 116.

Authors answer: Done as suggested.

Although is admirable your precision in describing the city of Junín, I suggest presenting table 2 in the supplementary section, since a very good explanation of why it is important to analyze this city is provided in section 2.2.

Authors answer: We express our gratitude to the reviewer for her/his kind words on our description of Junín. Please note, however, that without Table 2, the description is very incomplete, and the reader needs this information. We hope the reviewer will not disagree with us in having this table in the main text. We already have 3 Tables as supplementary files.

I suggest to describe the PF tests with more details with a subparagraph for each test, to make the methodology of this study reproducible in other studies. Also, I suggest to add reliability and IntraClass Correlation (ICC) for these tests.

Authors answer: Done as suggested. We extended our section 2.4 with information for all tests.

Was the shuttle run a 10 x 5m or 5x5m? Because in the EUROFIT tests battery there is the 10x5m shuttle run, while reading your description I understood that you used a 5x5m shuttle run. Please clarify.

Authors answer: We thank the reviewer for this call. More information was added in section 2.3.

I suggest to show table S2 after table 1 in the method section and not in the supplementary.

Authors answer: We thank the reviewer for this suggestion. Please note, however, that this implies that Tables S2 (boys) and S3 (girls) will be added to the main text. This will make this section too dense. Hoping that the reviewer will not disagree with us on this, and in order to keep this section as “neat as possible”, we will keep tables S2 and S3 as supplementary files.

Since BMIz-score is a crucial marker for health in childhood, I suggest adding the BMIz-score in table S2 and assessing the differences in BMIz-score between the two samples to strengthen your data.

Authors answer: We really appreciate the reviewer suggestion. Tables S1 and S2 only display adjusted means (height and weight as covariates) for all PF tests. Having BMIz in these tables would probably confuse the reader. Hence, and hoping the reviewer will not disagree with us, we will add this information in table S3. Please note, however, that we will only display means and standard deviations. In fact, physical growth trends in these Peruvian cohorts are now being investigated and will be reported, hopefully, in the near future.

Results: Please delete the term “significant” when reporting the results (for example line 203).

Authors answer: Done as suggested.

Please in the figure where there is a significant difference in PF between the age insert an appropriate symbol.

Authors answer: These figures were remade, and the reviewer’s suggestion is now present.

In line 219 the authors stated that the boys in 2019 were stronger than the boys in 2009. But in the table boys in 2019 jumped had a lower performance than their peers in 2009. This typo is also in lines 200-201 when reporting the results of the girls. Please correct.

Authors answer: Done as suggested.   

In Tables 3 and 4 I suggest including a row above the others explaining the gender of subjects in the table.

Authors answer: Done as suggested.

Discussion: In the discussion session the authors described an increasing trend in PF abilities in both sexes, but the table presented in the results section shows a decline (for example the standing broad jump in boys decreased in all the ages while in the discussion you said the opposite (Line 271). Accordingly, I suggest to modify the section of the discussion that compares these results with other studies.

Authors answer: We thank the reviewer for this suggestion, but we believe that there must be some misperception. In fact, with the exception of sit-and-reach, all other PF test results, within each sex, and within each cohort, increase with age.

Line 291–318 The explanation of the results obtained by the age-by-year analysis does not seem correlated with the results of the analysis. Specifically, you first stated that a possible explanation is due to the agility sprint, but since you compared children of the same age and found a difference between the two groups (2009-2019), your explanation is inadequate. Also, even if the increased hour of physical education in school could slightly improve the agility of children, this amelioration should have influenced children of all ages in the 2019 group and not only the performance of the 10- and 11-years old boys and girls.

Authors answer: We thank the reviewer for this comment, and we fully agree that our putative speculation may not explain our findings. Although we have cross-sectional data from a time-lag of 10 years, it is always very difficult, if not impossible, to infer causation even if in some epidemiological designs this may be possible (Savitz & Wellenius, 2022). Further, as we mentioned in our study limitations, we do not have any information on children’s daily physical activity and/or sports participation which could probably provide clues about possible links to the agility data. We re-phrased some parts of this explanation hoping that they may be in line with the reviewer concern.

Reference

Savitz DA, Wellenius GA. Can Cross-Sectional Studies Contribute to Causal Inference? It Depends. Am J Epidemiology. 2022 kwac037. Advance online publication. https://doi.org/10.1093/aje/kwac037

Moreover, I suggest at lines 293-294 clarify these results since you have not compared boys and girls together. Specifically, you have reported that “favoring children from 2009 at 6 and 7 years” but as reported in table 3 and 4, you found a decreased performance for the shuttle run only in the 7 years old boys and not in girls. Then you reported “children from 2019 at 8, 10 and 11 years” but the improved performance at 10 years old was showed only in girls, while in any of both sexes was found an improved performance at the age of 8. Please be careful when describing your results.

Authors answer: We thank the reviewer for this suggestion. We re-wrote parts of the text.

Finally, I suggest adding a brief section explaining the possible positive/negative repercussion of these trends on children’s health and how your results could be helpful for Junìn physical education teachers and trainers (for example focusing their activities on the decreased physical fitness abilities).

Authors answer: We thank the reviewer for this suggestion, and more information was added in the conclusion section.

Conclusions: For the same reasons stated in the discussion section please rephrase this paragraph.

Authors answer: Done as suggested.

Abstract: For the same reasons stated for the discussion section, I suggest updating this paragraph and add the correct interpretation of the results.

Authors answer: Done as suggested.